# Robust Prompt Optimization for Large Language Models Against Distribution Shifts

**Moxin Li[1], Wenjie Wang[1]\*, Fuli Feng[2, 3], Yixin Cao[4], Jizhi Zhang[2]**
**Tat-Seng Chua[1]**

[1]National University of Singapore, [2]University of Science and Technology of China
[3]Institute of Dataspace, Hefei, Anhui, China, [4]Singapore Management University
limoxin@u.nus.edu, wangwenjie@u.nus.edu, fulifeng93@gmail.com,
caoyixin2011@gmail.com, cdzhangjizhi@mail.ustc.edu.cn, dcscts@nus.edu.sg

## Abstract

Large Language Model (LLM) has demonstrated significant ability in various Natural Language Processing tasks. However, their effectiveness is highly dependent on the phrasing of the task prompt, leading to research on automatic prompt optimization using labeled task data. We reveal that these prompt optimization techniques are vulnerable to distribution shifts such as subpopulation shifts, which are common for LLMs in real-world scenarios such as customer reviews analysis. In this light, we propose a new problem of robust prompt optimization for LLMs against distribution shifts, which requires the prompt optimized over the labeled source group can simultaneously generalize to an unlabeled target group. To solve this problem, we propose Generalized Prompt Optimization framework, which incorporates the unlabeled data from the target group into prompt optimization. Extensive experimental results demonstrate the effectiveness of the proposed framework with significant performance improvement on the target group and comparable performance on the source group.

## 1 Introduction

LLMs have gained significant attention for their remarkable performance in a broad range of Natural Language Processing (NLP) tasks (Ouyang et al., 2022; Chung et al., 2022; Brown et al., 2020; Touvron et al., 2023). This success has led to a shift in the paradigm of solving NLP tasks, moving away from training task-specific deep models towards developing task-specific strategies to effectively utilize LLMs (Wei et al., 2022; Kojima et al., 2022; Wang et al., 2022a; Ye et al., 2023b). In the new paradigm, the prompt becomes a crucial factor in ensuring the effectiveness of LLM on the NLP task, since even slight variations in prompt phrasing can largely affect LLM output (Reynolds and

---
\*Corresponding author.

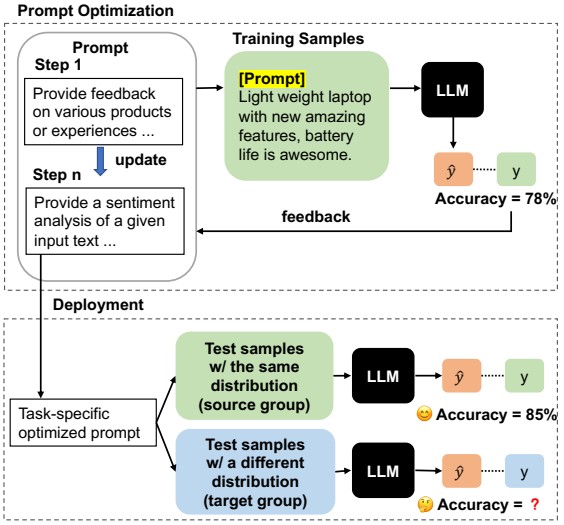

Figure 1: Illustration of prompt optimization under distribution shifts. Existing prompt optimization solutions aim to improve the LLM performance on the training data, while it is unclear whether the optimized prompt can be generalized to testing data of the same task but with distribution shifts.

McDonell, 2021; Gao et al., 2021), making prompt optimization a promising research direction.

Existing research has explored automatic prompt optimization methods to eliminate manual effort in identifying effective prompts for a given task. These methods can be gradient-based or gradient-free, depending on the availability of model gradients. Gradient-based methods optimize the prompt by calculating its gradients through the LLM (Schick and Schütze, 2021b,a; Hu et al., 2022). Gradient-free methods update prompts based on LLM outputs using techniques such as an iterative search-and-select over the prompt space (Zhou et al., 2023; Prasad et al., 2022; Pryzant et al., 2023). This work focuses on gradient-free prompt optimization as LLMs are evolving into black-box API services (Sun et al., 2022).

Current gradient-free prompt optimization methods ignore distribution shifts (Wang et al., 2023),

where the data an LLM serves may differ from the labeled data for prompt optimization. Real-world NLP applications often encounter distribution shifts, such as new user groups with distinct linguistic habits in customer review analysis. It is unclear if prompts hinder the robustness of LLMs against distribution shifts. To answer this question, we conduct experiments with the representative *gpt-3.5-turbo-0301* model and prompts optimized by APE (Zhou et al., 2023) over paired data groups with distribution shifts. Results on 30 pairs of data groups from six tasks show the risk of significant performance gaps under certain distribution shifts.

Based on this finding, we propose a new *robust prompt optimization* problem, which aims to optimize task-specific prompts with consideration of performance on both source and target groups under different distributions. Given an NLP task such as sentiment analysis, our problem setting has a labeled source group similar as the conventional prompt optimization setting and a unlabeled target group. We keep the target group unlabeled for the consideration that distribution shifts happen along time in practice. Labeling the newly coming target group will cause unnecessary labor cost and latency. Accordingly, the main challenge for solving this robust prompt optimization problem is incorporating unlabeled data into prompt optimization.

To this end, we propose the *Generalized Prompt Optimization* (GPO) framework to obtain a task-specific prompt for both source and target groups. To jointly considering the two groups in prompt optimization, the key lies in labeling the target group in an automatic and reliable manner by adapting knowledge from the labeled source group. Towards this goal, we leverage the strong power of LLM in zero-shot labeling, and prompt ensemble to enhance the labeling robustness. Experimental results on three tasks demonstrate the effectiveness of our framework in improving the performance on the target group and simultaneously preserving a comparable performance on the source group. To sum up, our contributions are threefold:

- We reveal the robustness issue of prompt optimization against distribution shifts and propose a new robust prompt optimization problem.

- We propose the Generalized Prompt Optimization framework, which generates robust prompts considering both labeled and unlabeled data.

- We conduct extensive experiments on three NLP

tasks, validating the rationality and effectiveness of our proposed framework.

## 2 Preliminary Experiments

**Prompt optimization** aims to find the best prompt $\mathbf{p}$ that can instruct LLMs to predict the output $\mathbf{y}$ based on the concatenation of $\mathbf{p}$ and task input $\mathbf{x}$, where $\mathbf{x}, \mathbf{y}$ and $\mathbf{p}$ are all sequences of tokens. Formally, given an NLP task with a dataset $\{(\mathbf{x}, \mathbf{y})\}$ following a distribution $P$, the goal is to obtain

$$\mathbf{p}^o = \arg\max_{\mathbf{p} \in \mathcal{Z}} \mathbb{E}_{(\mathbf{x},\mathbf{y}) \sim P}[r(\text{LLM}(\mathbf{p}, \mathbf{x}), \mathbf{y})], \quad (1)$$

where $\mathcal{Z}$ denotes the prompt optimization space and $r$ is the evaluation metric to compare the LLM output with the ground truth output $\mathbf{y}$, *e.g.*, Accuracy. Existing studies usually leverage gradient-based or gradient-free methods to automatically optimize the prompts. Since LLMs are evolving into black-box API services, gradient-free methods become increasingly important. However, they ignore distribution shifts between training and testing data. In this light, we conduct controlled experiments to answer the following research question:

> *Are prompts optimized by existing gradient-free methods robust to distribution shifts?*

### 2.1 Evaluation Protocol

We conduct the controlled experiments between a pair of data groups with distribution shifts, *i.e.*, a *source group* $\{(\mathbf{x}_s, \mathbf{y}_s)\}$ following a distribution $P_s$, and a *target group* $\{(\mathbf{x}_t, \mathbf{y}_t)\}$ with a distribution $P_t$, where $P_t \neq P_s$. We intend to examine whether the prompt $\mathbf{p}^s$ optimized on the source group can generalize to the target group. Specifically, given $\mathbf{p}^s$ and $\mathbf{p}^t$ optimized on the target group, we compare the performance of $\mathbf{p}^s$ on the target group $\mathbb{E}_{(\mathbf{x},\mathbf{y}) \sim P_t}[r(\text{LLM}(\mathbf{p}^s, \mathbf{x}), \mathbf{y})]$ with that of $\mathbf{p}^t$ $\mathbb{E}_{(\mathbf{x},\mathbf{y}) \sim P_t}[r(\text{LLM}(\mathbf{p}^t, \mathbf{x}), \mathbf{y})]$.

**Datasets**. We select 16 datasets from six popular NLP tasks, where each pair of groups under the same task is treated as the source and target groups. Following recent out-of-distribution (OOD) research (Yang et al., 2022), we take each dataset as a group and regard different backgrounds and topics across the datasets as the distribution shifts. For the sentiment analysis task, we adopt Yelp (Zhang et al., 2015), Flipkart (Vaghani and Thummar, 2023), IMDB (Maas et al., 2011) and Amazon (Zhang et al., 2015) of different topics.

| | Target MNLI | ANLI |
|---|---|---|
| Source | | |
| MNLI | $73.4 \pm 1.0$ | $45.4 \pm 1.9$ |
| ANLI | $73.3 \pm 1.3$ | $46.0 \pm 1.5$ |

(a) Natural language inference

| | Target RTE | HANS |
|---|---|---|
| Source | | |
| RTE | $78.3 \pm 0.8$ | $67.2 \pm 1.1$ |
| HANS | $79.0 \pm 0.8$ | $68.4 \pm 1.8$ |

(b) Textual entailment

| | Target DSTC7 | Ubuntu Dialog | MuTual |
|---|---|---|---|
| Source | | | |
| DSTC7 | $58.4 \pm 0.8$ | $78.9 \pm 0.3$ | $74.2 \pm 2.2$ |
| Ubuntu Dialog | $56.9 \pm 1.3$ | $78.7 \pm 0.5$ | $74.4 \pm 2.1$ |
| MuTual | $52.2 \pm 4.4$ | $74.7 \pm 6.0$ | $76.7 \pm 3.4$ |

(c) Dialog

Table 1: Results for tasks without large generalization performance gap across groups.

| | Target Yelp | Flipkart | IMDB | Amazon |
|---|---|---|---|---|
| Source | | | | |
| Yelp | $\mathbf{79.7 \pm 0.7}$ | $78.4 \pm 1.9$ | $87.1 \pm 1.9$ | $88.4 \pm 1.9$ |
| Flipkart | $69.1 \pm 8.7$ | $\mathbf{85.1 \pm 2.9}$ | $85.2 \pm 9.4$ | $85.9 \pm 12.5$ |
| IMDB | $71.1 \pm 8.2$ | $76.9 \pm 13.4$ | $\mathbf{91.9 \pm 0.9}$ | $90.4 \pm 5.2$ |
| Amazon | $75.5 \pm 1.5$ | $\mathbf{85.6 \pm 2.1}$ | $91.5 \pm 0.8$ | $\mathbf{93.5 \pm 1.4}$ |

(a) Sentiment analysis

| | Target SocialIQA | PIQA | OpenbookQA |
|---|---|---|---|
| Source | | | |
| SocialIQA | $75.6 \pm 1.4$ | $82.0 \pm 6.0$ | $71.2 \pm 5.2$ |
| PIQA | $68.9 \pm 6.9$ | $83.6 \pm 2.9$ | $69.2 \pm 5.1$ |
| OpenbookQA | $\mathbf{79.9 \pm 1.0}$ | $\mathbf{84.5 \pm 1.6}$ | $\mathbf{80.1 \pm 2.4}$ |

(b) Commonsense QA

| | Target Number | Spans |
|---|---|---|
| Source | | |
| Number | $51.9 \pm 2.8$ | $20.1 \pm 1.3$ |
| Spans | $\mathbf{57.7 \pm 2.9}$ | $\mathbf{63.1 \pm 2.2}$ |

(c) DROP

Table 2: Results for tasks with significant generalization performance gap across groups. Bold font indicates the largest value for each column.

For the natural language inference task, we utilize MNLI (Williams et al., 2018), and ANLI (Nie et al., 2020) which is an adversarial dataset for MNLI. For the textual entailment, we use RTE (Wang et al., 2018) and its OOD dataset HANS (McCoy et al., 2019). For commonsense QA, we use SocialIQA (Sap et al., 2019), PIQA (Bisk et al., 2020), and OpenbookQA (Mihaylov et al., 2018), which focus on different types of commonsense knowledge. For the multi-turn dialog reasoning, we use DSTC7 (Gunasekara et al., 2019), Ubuntu Dialog (Lowe et al., 2015), and MuTual (Cui et al., 2020). Besides, for the numerical QA task, we use the samples of two different answer types (*i.e.,* numerical values and text spans) in DROP (Dua et al., 2019) as two groups. See Appendix A.1 for details.

**Experimental Setup**. We adopt APE (Zhou et al., 2023), an effective gradient-free prompt optimization method, for prompt generalization analysis. To highlight the effect of prompts, we conduct experiments under the zero-shot setting without in-context examples. For the backbone LLMs, we leverage *gpt-3.5-turbo-0301* by calling the OpenAI API[1]. For all classification tasks (all tasks except for DROP), we use accuracy as the evaluation metric. For DROP, we utilize its standard evaluation metric — F1. Following the setting of APE, we randomly sample $N$-shot training and $N$-shot validation samples for prompt optimization, and repeat the experiments for five runs with different sampled

data to report the averaged results. More implementation details can be found in Appendix A.2.

## 2.2 Experimental Results

**Demonstration of Generalization Performance Gap.** Table 1 shows the tasks without a large generalization gap between the performance of prompts $\mathbf{p}^s$ and $\mathbf{p}^t$, and Table 2 shows the tasks with large gaps (Accuracy gap>8.0) on some groups. The row headers refer to the source groups for prompt optimization while the column headers show the target groups to test optimized prompts. The generalization performance gap between $\mathbf{p}^s$ and $\mathbf{p}^t$ can be observed by comparing the values in the same column.

From the tables, we can observe: 1) The generalization performance gap may not exist for previously studied OOD and adversarial groups (see Table 1), including the groups of the natural language inference and the textual entailment tasks. This is possibly attributed to the strong generalization ability of LLMs. 2) However, under some data groups of Table 2 such as the sentiment analysis datasets (*e.g.,* Flipkart and Yelp) and the commonsense QA datasets with different topics (*e.g.,* PIQA and OpenbookQA), and the DROP groups with different answer types, there are still significant generalization performance gaps, demonstrating the existence of the generalization issue of prompt optimization. 3) Surprisingly, the prompt $\mathbf{p}^s$ op-

---

[1] https://chat.openai.com/.

| Source \ Target | Yelp | Flipkart | IMDB | Amazon |
|---|---|---|---|---|
| Yelp | - | 0.33 | 1.62 | 1.62 |
| Flipkart | 0.30 | - | 0.57 | 0.56 |
| IMDB | **0.25** | 0.29 | - | **0** |
| Amazon | **0.25** | **0.27** | **0** | - |

(a) Label distribution shifts. Smaller values indicate less distribution shifts.

| Source \ Target | Yelp | Flipkart | IMDB | Amazon |
|---|---|---|---|---|
| Yelp | - | 0.65 | 0.73 | 0.76 |
| Flipkart | 0.59 | - | 0.55 | 0.63 |
| IMDB | 0.70 | 0.63 | - | **0.81** |
| Amazon | **0.71** | **0.70** | **0.78** | - |

(b) Input similarity. Larger values indicate less distribution shifts.

Table 3: Results for (a) label distribution shifts (b) input similarity of the sentiment analysis datasets. Bold font indicates the least distribution shift for each column.

| | SocialIQA | PIQA | OpenbookQA |
|---|---|---|---|
| word 1-gram | 0.43 | 0.51 | **0.58** |
| char 4-gram | 0.50 | 0.60 | **0.65** |

(a) The n-gram diversity.

| Source \ Target | SocialIQA | PIQA | OpenbookQA |
|---|---|---|---|
| SocialIQA | - | 0.39 | 0.38 |
| PIQA | 0.47 | - | **0.46** |
| OpenbookQA | **0.51** | **0.52** | - |

(b) The word 1-gram coverage ratio between groups.

| Source \ Target | SocialIQA | PIQA | OpenbookQA |
|---|---|---|---|
| SocialIQA | - | 0.51 | 0.51 |
| PIQA | 0.60 | - | **0.58** |
| OpenbookQA | **0.66** | **0.64** | - |

(c) The character 4-gram coverage ratio between groups.

Table 4: Evaluation on (a) the n-gram diversity and (b) word 1-gram coverage ratio (c) character 4-gram coverage ratio of commonsense QA datasets to study the even higher generalization performance. Bold font indicates the largest value for each column.

timized from the source group does not always perform worse than the prompt $\mathbf{p}^t$ optimized on the target group. In Table 2(b), $\mathbf{p}^s$ from OpenbookQA performs even better than $\mathbf{p}^t$ for SocialIQA. Besides, for DROP in Table 2(c), $\mathbf{p}^s$ from Spans also performs better than $\mathbf{p}^t$ from Number. In the following section, we try to explore the reasons for the above three observations.

**Exploration on the Factors Affecting Prompt Robustness.** Based on the above observations, we further explore two research questions.
**Q1**: *Why do the prompts optimized on source groups perform differently on a target group?*
**Q2**: *Why does the prompt optimized on the source group perform even better than the prompt optimized on the target group in some cases?*

For Q1, we conjecture that the varied performance gaps are attributed to different distribution shifts between the source and target groups. To verify this, we examine two metrics to measure two kinds of distribution shifts: 1) the label shifts measured by the KL divergence, and 2) the input similarity quantified by the n-gram similarity of the input corpora of the two groups. Their detailed implementation is illustrated in Appendix A.3. We show the results of the sentiment analysis task as an example in Table 3. We can observe that the smallest label distribution shifts and the largest input similarity in Table 3 generally coincide with the best generalization performance on each target group in Table 2, indicating the correlation between distribution shifts and generalization per-

formance. Nevertheless, the two metrics cannot perfectly explain the performance on all tasks (*cf.* Appendix A.3). Therefore, Q1 is still a challenging research question, requiring further exploration in future work.

For Q2, we conjecture that the outstanding generalization performance is because a source group with large diversity covers heterogeneous patterns in the target group, leading to a more robust prompt $\mathbf{p}^s$ than $\mathbf{p}^t$. To explore this, we measure the heterogeneity of source and target groups by calculating the percentage of unique n-grams, and the percentage of n-grams of the target group covered by the source group. For illustration, we present the results of the commonsense QA task in Table 4. From Table 4(a), we can observe that OpenbookQA has the most diverse input according to the n-gram statistics. Moreover, OpenbookQA covers a large proportion of n-grams of SocialIQA and PIQA. These partly explain the superiority of the prompts optimized on OpenbookQA (see Table 2).

## 3 Robust Prompt Optimization

In this section, we first formulate a robust prompt optimization problem and propose a GPO framework to enhance the robustness of the prompts.

### 3.1 Problem Definition

To enhance the generalization ability of prompts, we propose a robust prompt optimization prob-

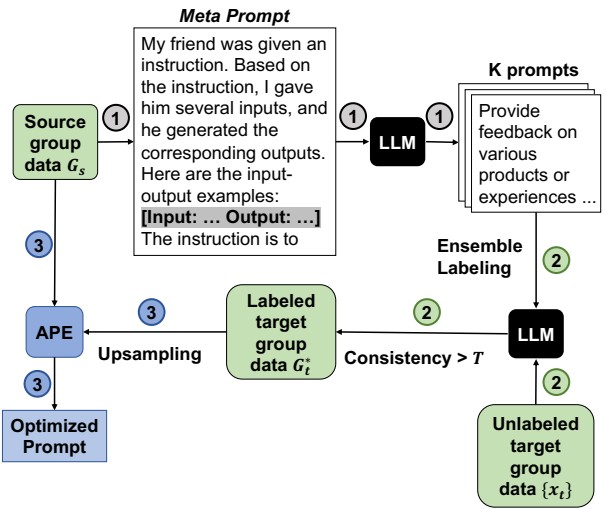

Figure 2: The GPO Framework.

lem. Specifically, given an NLP task such as sentiment analysis, it aims to optimize a task-specific prompt for the data groups with different distributions. We consider the popular scenario where a source group $G_s = \{(\mathbf{x}_s, \mathbf{y}_s)\}$ following a distribution $P_s$ and $\{\mathbf{x}_t\}$ in a unlabeled target group $G_t = \{(\mathbf{x}_t, \mathbf{y}_t)\} \sim P_t \ (P_t \neq P_s)$ are available while $\{\mathbf{y}_t\}$ is unseen during prompt optimization. The objective becomes utilizing $G_s = \{(\mathbf{x}_s, \mathbf{y}_s)\}$ and $\{\mathbf{x}_t\}$ to optimize a task-specific prompt robust to the samples from either $P_s$ or $P_t$.

**Reasons for Access to Unlabeled Target Group.** In a real-world deployment, LLMs continually encounter the testing data with distribution shifts. Collecting the input features $\{\mathbf{x}_t\}$ of the target group is feasible. For example, when using LLMs as web services to solve user queries of certain NLP tasks, it is easy to collect extensive user queries as unlabeled target groups. However, labeling $\{\mathbf{x}_t\}$ may be time-consuming and costly, and thus we intend to optimize robust prompts without the labels of the target group.

**A Task-Specific Prompt vs. One Prompt for Each Group.** To tackle the generalization issue of optimized prompts, an intuitive approach is to optimize a separate prompt for each data group, yet this simplistic approach faces several limitations in real scenarios. In real-world deployment, it not only requires additional computation costs to construct more prompts, but also needs to accurately classify each testing sample into the appropriate group of the same distribution, thereby resulting in increased computation costs, latency, and new challenges for precise group classification. Further-

more, the collected source group data cannot cover all potential target groups, and the prompts optimized on the source groups may inevitably test on the examples from previously unseen groups. Thus, we aim at improving the generalization ability of one task-specific prompt across different groups.

## 3.2 GPO Framework

To obtain a robust prompt for both the source and target groups, it is natural to jointly consider $G_s$ and $G_t$ for prompt optimization. However, $G_t$ lacks the labels $\{\mathbf{y}_t\}$ that are commonly required by the gradient-free optimization methods (refer to Table 5 for the inferior results without labeling). With the impressive capabilities of LLMs on zero-shot labeling, we propose to utilize LLMs to label $\{\mathbf{x}_t\}$. Considering that noisy labels may damage the quality of optimized prompts, we further present two strategies to improve labeling accuracy.

As illustrated in Figure 2, we first propose a *Meta Prompt* to instruct LLMs to acquire knowledge from the labeled source group and generate a series of prompts. Thereafter, we utilize a prompt ensemble labeling strategy to apply generated prompts to an LLM for precise labeling of $\{\mathbf{x}_t\}$. In detail, we derive a three-step framework to perform the labeling with two strategies, and then conduct joint prompt optimization as shown in Figure 2.

1. **Prompt Generation via Meta Prompt**. Following APE, we utilize a Meta Prompt to ask LLM to generate prompts for labeling by feeding the examples of $G_s$ (see an example in Figure 2). Based on strong language understanding and reasoning abilities, LLMs can infer the relationships between the inputs and outputs of the examples and provide general and precise task prompts. We use different splits of $G_s$ to generate $K$ different prompts in total.

2. **Prompt Ensemble Labeling Strategy.** Given $K$ prompts, we utilize each of them to label $\{\mathbf{x}_t\}$ with an LLM, and thus obtain $K$ candidate labels for each example. We adopt an ensembling strategy and select the label with the highest consistency among the $K$ candidate labels for each example. Besides, inspired from Wang et al. (2022a), we set a consistency threshold $T \in [0, 1]$ to only accept the labeled examples that have more than $T$ percent of prompts agreed on the label. Eventually, we obtain a filtered labeled set $G_t^*$ for the target group.

3. **Joint Prompt Optimization.** Finally, we mix $G_s$ and $G_t^*$ to run APE for joint prompt optimization and obtain the final optimized prompt. As $G_t^*$ may have fewer samples than $G_s$ after filtering with $T$, we perform a random upsampling on $G_t^*$ to have the same data number as $G_s$ before running APE. A brief illustration about APE can be found in Appendix A.2.

## 4 Experiments

### 4.1 Setup

**Datasets.** We experiment GPO with three tasks: sentiment analysis, commonsense QA, and DROP. For each task, we select a pair of groups with generalization performance gap as source and target groups, and ablate the labels for the target groups.

**Compared Methods.** We adopt the following baseline methods: 1) APE; 2) APO (Pryzant et al., 2023), the state-of-the-art gradient-free prompt optimization method for LLM; 3) APE-ut, a naive generalization solution by incorporating the unlabeled target group input into APE; 4) the Upper Bound, which represents the performance of the prompt optimized on the target group data with ground-truth labels by APE; and 5) our proposed GPO; We also show the results of simple human-written prompts that are general for the task, and the revised versions by PromptPerfect[2] which is an automatic prompt engineering website.

**Evaluation Protocol.** We utilize two strategies for testing: Top 1 and Ensemble. Top 1 refers to using the single optimized prompt with the best validation performance, while Ensemble refers to labeling with all obtained $K$ prompts and accept the output with the most agreement on the prompts. We utilize the same $N$-shot data as the preliminary experiments and also report the averaged results for five runs. More implementation details are illustrated in Appendix A.4.

### 4.2 Performance Comparison

**Compare to Generated Prompts.** From Table 5, we can observe the followings: 1) GPO achieves superior performance for all target groups in both Top 1 and Ensemble testing, validating its effectiveness. However, there is still space for improvement towards the Upper Bound for all tasks, showing the challenge of the generalization problem. 2) GPO achieves comparable source group performance for all tasks, showing its improvement on the target

| | Yelp (Source) | | Flipkart (Target) | |
|---|---|---|---|---|
| | Top 1 | Ensemble | Top 1 | Ensemble |
| APE | **79.7 ± 0.7** | **79.7 ± 1.0** | 78.4 ± 1.9 | 81.3 ± 1.4 |
| APO | 78.9 ± 0.5 | **79.7 ± 0.8** | 74.7 ± 3.0 | 76.4 ± 1.4 |
| APE+ut | 78.9 ± 1.4 | 78.8 ± 1.4 | 80.3 ± 2.0 | 80.7 ± 2.1 |
| GPO | 79.1 ± 0.7 | 78.7 ± 0.9 | **80.5 ± 2.1** | **84.5 ± 2.0** |
| Upper Bound | - | - | 85.1 ± 2.9 | 87.2 ± 0.5 |

(a) Sentiment analysis.

| | SocialIQA (Source) | | OpenbookQA (Target) | |
|---|---|---|---|---|
| | Top 1 | Ensemble | Top 1 | Ensemble |
| APE | 75.6 ± 1.4 | 69.6 ± 5.3 | 71.2 ± 5.2 | 74.8 ± 3.2 |
| APO | 76.1 ± 2.7 | 72.3 ± 2.6 | 72.4 ± 2.5 | 66.1 ± 7.2 |
| APE+ut | **77.9 ± 1.3** | **78.9 ± 0.8** | 77.5 ± 3.0 | 79.2 ±1.2 |
| GPO | 76.7 ± 2.0 | **78.9 ± 1.2** | **78.7 ± 3.3** | **79.7 ± 0.8** |
| Upper Bound | - | - | 80.1 ± 2.4 | 80.8 ± 1.1 |

(b) Commonsense QA.

| | Number (Source) | | Spans (Target) | |
|---|---|---|---|---|
| | Top 1 | Ensemble | Top 1 | Ensemble |
| APE | 51.9 ± 2.8 | 51.0 ± 3.2 | 20.1 ± 1.3 | 18.2 ± 0.2 |
| APO | **55.7 ± 0.8** | **54.5 ± 2.1** | 20.2 ± 2.4 | 20.0 ± 2.2 |
| APE+ut | 52.0 ± 1.8 | 53.1 ± 1.2 | 16.1 ± 3.5 | 17.7 ± 2.8 |
| GPO | 52.2 ± 6.0 | 53.6 ± 3.0 | **27.7 ± 12.0** | **26.7 ± 4.9** |
| Upper Bound | - | - | 63.1 ± 2.2 | 63.7 ± 0.8 |

(c) DROP.

Table 5: Results of the compared methods. Bold font indicates the best performance for each column.

group does not largely hinder the source group. Compared with APE, GPO shows increased performance on the source groups of SocialIQA and Number by incorporating the target group data, which is in line with the finding in Table 2. 3) Across baselines, APO outperforms APE on the source groups of the last two tasks and achieve comparable performance on sentiment analysis, showing its effectiveness for prompt optimization. However, the generalization ability is only comparable to APE since APO performs worse than APE on several target groups. 4) APE-ut achieves improved target group performance for the first two task, indicating the benefit of incorporating unlabeled target group data for generalization. However, for Spans where obtaining accurate target labels is challenging (as shown by the low F1 values), APE-ut largely underperforms GPO, showing the importance of target group labeling especially for difficult tasks.

**Compare to Human-written Prompts.** From Table 6, we further observe that GPO outperforms human-written prompts and PromptPerfect for sentiment analysis and commonsense QA tasks. However, on the most difficult task DROP, GPO underperforms human-written prompts. This is poten-

|  | Yelp (Source) | Flipkart (Target) | SocialIQA (Source) | OpenbookQA (Target) | Number (Source) | Spans (Target) |
|---|---|---|---|---|---|---|
| Human | **78.7** | 80.0 | 71.3 | 60.0 | **54.9** | **37.1** |
| PromptPerfect | 77.3 | 83.3 | 74.7 | 64.0 | 54.0 | 26.9 |
| GPO best | **78.7** | **84.5** | **78.9** | **79.7** | 52.2 | 27.7 |

Table 6: Performance comparison for the human-written prompts, PromptPerfect and the more effect testing strategy of GPO (Top 1 or Ensemble, denoted as GPO best). Bold font indicates the best performance for each column.

tially because the inaccurate labels for Spans hinder the prompt optimization. Similarly, PromptPerfect also fail to optimize human-written prompts for DROP.

## 4.3  Ablation Study

|  | Yelp | | Flipkart | |
|---|---|---|---|---|
|  | Top 1 | Ensemble | Top 1 | Ensemble |
| GPO | 79.1 ± 0.7 | 78.7 ± 0.9 | 80.5 ± 2.1 | **84.5 ± 2.0** |
| w/o cons | 78.8 ± 1.2 | **78.7 ± 0.4** | **81.5 ± 1.4** | 84.0 ± 0.9 |
| w/o cons+t-train | **79.9 ± 0.8** | **79.7 ± 1.0** | 80.3 ± 3.2 | 81.3 ± 1.4 |

(a) Sentiment analysis.

|  | SocialIQA | | OpenbookQA | |
|---|---|---|---|---|
|  | Top 1 | Ensemble | Top 1 | Ensemble |
| GPO | 76.7 ± 2.0 | **78.9 ± 1.2** | **78.7 ± 3.3** | **79.7 ± 0.8** |
| w/o cons | 76.0 ± 2.8 | 78.1 ± 1.4 | 77.6 ± 3.8 | 78.8 ± 2.2 |
| w/o cons+t-train | **77.9 ± 1.6** | 69.6 ± 5.3 | 78.2 ± 2.2 | 74.8 ± 3.2 |

(b) Commonsense QA.

|  | Number | | Spans | |
|---|---|---|---|---|
|  | Top 1 | Ensemble | Top 1 | Ensemble |
| GPO | **52.2 ± 6.0** | **53.6 ± 3.0** | **27.7 ± 12.0** | **26.7 ± 4.9** |
| w/o cons | 49.3 ± 2.8 | 51.0 ± 2.1 | 20.6 ± 2.1 | 22.2 ± 3.2 |
| w/o cons+t-train | 51.3 ± 3.6 | 50.9 ± 1.6 | 20.4 ± 1.9 | 18.7 ± 2.2 |

(c) DROP.

Table 7: Ablation study. Bold-font and underline indicate the best and second-best results, respectively.

We study the effect of prompt ensemble labeling and joint prompt optimization by evaluating two modifications of GPO: (1) setting the consistency threshold as 0, denoted as *w/o cons*; and (2) removing the target group training data during the final prompt generation, denoted as *w/o cons+t-train*. From Table 7, we can observe that: 1) In all cases except for Flipkart with Top 1 evaluation, GPO performs better than *w/o cons* on target groups, showing the effectiveness of the consistency threshold. 2) Among the three tasks, DROP has large improvement between *w/o cons* and GPO on both source and target groups then the other two tasks. We hypothesis that this discrepancy is related to the different degrees of improvement in the labeling accuracy by the consistency threshold, which will be further discussed in Section 4.4. 3) Comparing

|  | Flipkart | OpenbookQA | Spans |
|---|---|---|---|
| *w/o cons* | 81.9 | 69.8 | 3.6 |
| GPO | 94.2 | 84.3 | 3.7 |

Table 8: The labeling accuracy comparison for the target group training and validation data on GPO and *w/o cons*. The results for Spans here is accuracy instead of F1.

*w/o cons* and *w/o cons+t-train*, removing the target group training data benefits the Top 1 results of the source group, but harms the Ensemble results of the target groups. It has less effect on the target group Top 1 results since the two methods still use target group validation data.

## 4.4  In-depth Analysis

**Analysis on the Effect of the Consistency Threshold.** To further reveal the effect of consistency threshold, we first show the labeling accuracy of the target group training and validation data for GPO and *w/o cons* in Table 8. We can observe that applying the consistency threshold can improve the labeling accuracy for all target groups. By examining the relationship between this labeling accuracy improvement and the performance difference between GPO and *w/o cons* in Table 7, it can be explained that for Flipkart and OpenbookQA, where the labeling accuracy is already high under *w/o cons*, further improving the labeling accuracy by the consistency threshold is unlikely to achieve large performance gain. Conversely, in the case of Spans with low labeling accuracy, even a minor improvement can result in significant performance gains. To explore the connection between labeling accuracy and target group performance further, we conducted an experiment where we manually assigned incorrect labels to varying proportions (0%, 50%, and 90%) of the target training and validation data. The results are illustrated in Figure 3. It can be observed that as the percentage of incorrect labels increases, the overall performance on the target group generally decreases, emphasizing the importance of labeling accuracy for achieving effective generalization.

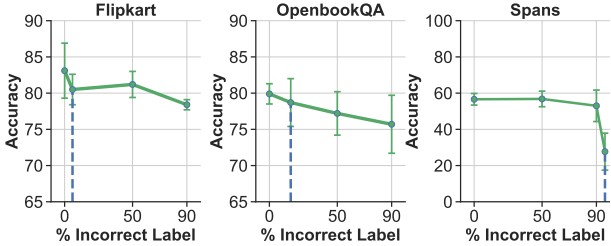

Figure 3: Target group performance under different percentage of wrong labels. The blue dotted line indicates the labeling accuracy of GPO as in Table 8.

| | Top 1 | | Ensemble | |
|---|---|---|---|---|
| | APE | GPO | APE | GPO |
| Vicuna-7B | 38.4 ± 25.3 | 63.5 ± 15.6 | 43.9 ± 21.3 | 71.9 ± 13.1 |
| Vicuna-13B | 66.8 ± 18.4 | 68.3 ± 13.7 | 60.7 ± 9.5 | 70.7 ± 10.8 |
| GPT-3.5 | **78.4 ± 1.9** | 80.5 ± 2.1 | 81.3 ± 1.4 | 84.5 ± 2.0 |
| GPT-4 | 77.5 ± 13.7 | **85.3 ± 2.7** | 83.3 ± 0.0 | **85.4 ± 2.4** |

Table 9: Performance comparison of APE and GPO on Flipkart of different backbone LLMs.

**GPO with Different Backbone LLMs.** We also conducted experiments with GPO using different backbone LLMs, including Vicuna 7B and 13B (Chiang et al., 2023) which are notable smaller-sized LLMs, and GPT-4 (OpenAI, 2023). Table 9 shows the generalization results on Flipkart with Yelp as the source group for APE and GPO on different backbone LLMs. Due to the small sizes of the Vicuna models, generating the exact sentiment label as the answer can be challenging. Therefore, we extract the sentiment labels from their outputs before calculating the accuracy. The results show that there is room for enhancing the generalization performance in APE across various LLMs, and GPO consistently outperforms APE in all cases. Notably, when applying GPO to the smaller Vicuna 7B model, there is a significant improvement that allows it to reach the same performance level as the Vicuna 13B model. Across LLMs, the smaller-sized Vicuna models achieve relatively worse performance, and the powerful GPT-4 achieves the best performance on GPO.

## 5 Related Work

**Generalization Ability and Robustness of LLM.**
Researchers have been investigating the generalization ability and robustness of LLMs since their recent breakthrough. LLMs like ChatGPT have shown significant improvement in out-of-distribution (OOD) and adversarial tasks (Wang et al., 2023), although they are still imperfect (Chen et al., 2023). Some LLMs still rely on shortcuts

and spurious correlation (Tang et al., 2023; Stolfo et al., 2022). Moreover, LLMs remain vulnerable to adversarial perturbations and achieve inconsistent results (Wang et al., 2023; Ye et al., 2023a; Liang et al., 2022). Additionally, LLMs demonstrate high sensitivity to the prompt (Reynolds and McDonell, 2021; Zhu et al., 2023) and the selection of in-context examples (Liu et al., 2022; Rubin et al., 2022). Lastly, instruction tuning allows LLMs to generalize to novel tasks (Ouyang et al., 2022; Wang et al., 2022b,a). We specifically focus on the generalization issue of prompt optimization on the distribution shifts within one task.

**Prompt Optimization.** Obtaining effective prompts for applying LLM in NLP tasks is a popular research area. Prompt tuning methods (Li and Liang, 2021; Lester et al., 2021; Qin and Eisner, 2021; Gu et al., 2022) learn soft continuous vectors as prompts in the LLM input using gradients from the task objective. Recent studies have also focused on gradient-free prompt optimization for black-box LLM, such as reinforcement learning-based methods (Zhang et al., 2023; Deng et al., 2022; Diao et al., 2022), search-based methods (Brown et al., 2020; Prasad et al., 2022; Pryzant et al., 2023), and other gradient-free optimization techniques like evolutionary algorithms (Sun et al., 2022) and boosting (Hou et al., 2022). Among them, the state-of-the-art methods leverage the power of LLMs for prompt optimization, such as prompt generation and evaluation by LLM (APE (Zhou et al., 2023)) and prompt editing following critiques (APO (Pryzant et al., 2023)), where we mainly compare with them. Notably, while some previous work on prompt tuning has addressed generalization across tasks and models (Su et al., 2022; Vu et al., 2021; Qin et al., 2023), and domain adaptation (Tam et al., 2022; Guo et al., 2022), this paper specifically focuses on the generalization issue of gradient-free prompt optimization.

## 6 Conclusion

In this paper, we revealed the generalization issue of prompt optimization for LLMs under distribution shifts. We observed that the prompt optimized on the source data group may have a performance drop on the target group with distribution shifts. We performed an initial analysis aiming at identifying the factors that correlate to the varied generalization performance across groups, including label distribution shift and input distribution sim-

ilarity. To enhance the generalization ability of LLMs, we proposed a Generalized Prompt Optimization framework to jointly consider the source and target groups for robust prompt optimization. Experimental results validated the effectiveness of our proposed framework in boosting the robustness of the prompts on the source and target groups. In future work, we plan to study the prompt generalization to unseen target groups without available inputs $\{\mathbf{x}_t\}$, and explore prompt generalization ability with in-context examples from different groups.

## Limitations

Firstly, this work discusses the generalization ability of prompts while ignoring the effect of other LLM inputs such as in-context examples. The choice of in-context examples might also affect the robustness of LLMs. Future work can look into the generalization issue of the prompt in combination with in-context examples. Secondly, this work assumes the availability of the inputs $\{\mathbf{x}_t\}$ of the target group. It is under-explored how to achieve generalized prompt optimization to completely unseen groups without $\{\mathbf{x}_t\}$. To improve the robustness on these groups, we believe it is helpful to extend this work toward robust prompt optimization on multiple heterogeneous groups. Thirdly, we acknowledge that the scope of our research is limited to black-box LLMs capable of understanding instructions, where gradient-free prompt optimization with instructing LLM is a suitable choice. For smaller LMs without instruction understanding abilities, *e.g.,* BERT (Devlin et al., 2019) and T5 (Raffel et al., 2020), they are generally not black-box and are more advantageous to utilize gradient-based prompt optimization methods.

## Acknowledgements

This work is supported by NExT Research Center, and the National Natural Science Foundation of China (62272437). We thank the reviewers for their constructive feedback.

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

# A Appendix

## A.1 Dataset Details

For each dataset, we use the original training set to split into training and validation sets, and randomly sample a subset from the original validation set as our test set as sometimes the labels for the original test set are not available. Following the official implementation of APE [3], we split the original training set with 1000 training samples, and the rest as validation samples. For MNLI, we sample the same number of matched and mismatched validation data as the test set. For ANLI, we use R2. For Yelp and Flipkart, we assign the review scores of 0 and 1 as negative, 3 as neutral, and 4, 5 as positive. For multi-turn dialog reasoning, we select the instances of MuTual within 5 dialog turns, Ubuntu and DSTC7 within 7 dialog turns, and reduce the number of choices to 4 for all three datasets. We show an example of LLM input for each task in Table 11, and the dataset statistics in Table 10.

## A.2 Additional Implementation Details for Preliminary Experiments

The APE performs prompt optimization by iteratively generating and selecting the prompts leveraging LLM. For prompt generation, it utilizes a meta prompt to instruct LLM to infer prompts from given input-output examples. Then, the generated prompts are evaluated on validation data to select the prompts with good task performance. After that, APE leverages LLM to perform Monte Carlo search by iteratively paraphrasing the current effective prompts and performing evaluation on them to obtain optimized prompts.

Following the official implementation, for prompt generation, the sampled $N$-shot training data are divided into $K$ splits to generate $K$

|  | # Train&Val | # Test | $N$ Shot | $K$ Prompt |
|---|---|---|---|---|
| Yelp | 650000 | 150 | 36 | 6 |
| Flipkart | 75138 |  |  |  |
| IMDB | 25000 |  |  |  |
| Amazon | 100000 |  |  |  |
| SocialIQA | 33410 | 150 | 36 | 6 |
| PIQA | 16113 |  |  |  |
| OpenbookQA | 4957 |  |  |  |
| Number | 2000 | 150 | 36 | 6 |
| Spans | 2000 |  |  |  |
| MNLI | 392702 | 1000 | 16 | 4 |
| ANLI | 45460 |  |  |  |
| RTE | 2490 | 277 | 16 | 4 |
| HANS | 30000 | 1000 |  |  |
| DSTC7 | 43824 | 150 | 9 | 9 |
| Ubuntu Dialog | 94107 |  |  |  |
| MuTual | 4783 |  |  |  |

Table 10: Statistics for the train, validation and test splits for each dataset, and the values of shot number $N$ and prompt number $K$ for each task. The Train&Val are further split into 1000 training samples and the rest as validation samples.

prompts by LLM for further selection. For each task, we try the value of $N$ as 9, 16, 25, 36, and $K$ as $N$'s factors, to ensure obtaining effective prompts, where APE is not very parameter sensitive. Moreover, we ablate the Monte Carlo search since it is optional and not significant for our tasks.

Given the randomness of the backbone LLM, we set the temperature of the LLM as 0, top p as 1.0. We set the max tokens for prompt generation as 100 to try to ensure no truncation, and keep other LLM parameters the same as the official APE implementation. The parameters $N$ and $K$ are shown in Table 10.

## A.3 Additional Details and Results for the Exploration on the Factors Affecting Prompt Robustness.

**Calculation of Q1 Metrics.** The **label distribution shift** quantifies the divergence of the label distributions between two groups for classification tasks, calculated by the KL divergence of their label distributions,

$$D_{KL} = \sum_{y \in \mathcal{Y}} Pr_s(y) log(\frac{Pr_s(y)}{Pr_t(y)})$$

where $\mathcal{Y}$ is the label space of the task, and $Pr_s(y)$ and $Pr_t(y)$ denote the probability of the label $y$ in the source and target groups, respectively.

[3] https://github.com/keirp/automatic_prompt_engineer/tree/main.

| Dataset | Input Example | Labels |
|---|---|---|
| Yelp | Dr. Goldberg offers everything I look for in a general practitioner. He's nice and easy to talk to without being patronizing; he's always on time in seeing his patients... | positive, negative, neutral |
| OpenbookQA | The sun is responsible for (A) puppies learning new tricks (B) children growing up and getting old (C) flowers wilting in a vase (D) plants sprouting, blooming and wilting. | A, B, C, D |
| MNLI | Premise: One of our number will carry out your instructions minutely. Hypothesis: A member of my team will execute your orders with immense precision. | entailment, neutral, contra-diction |
| HANS | Sentence 1: The doctors supported the scientist. Sentence 2: The scientist supported the doctors. | entailment, non−entailment |
| DSTC7 | S: Hello! A: Hello! S: I'm wondering for next semester what class should I take. A: Given your experience, I suggest you take EECS 280. S: Do you know what the size of that class is? Answer Choices: (A) EECS 481 covers dealing with structuring principles, pragmatic aspects of the production of software systems, design methodologies and informal analysis. (B) The class size is normally around 167 students. (C) Based on the classes you've taken, this class shouldn't be extremely demanding. (D) This course has a discussion section. | A, B, C, D |
| Number | Question: How many in percent weren't 45 to 64? Context: In the city, the year 2010 population was spread out with 26.3% under the age of 18, 13.6% from 18 to 24, 30.7% from 25 to 44, 21.1% from 45 to 64, and 7.2% who were 65 years of age or older. The median age was 32 years. For every 100 females, there were 92.5 males. For every 100 females age 18 and over, there were 88.4 males. | *e.g.,* 78.9 |

Table 11: Dataset examples for each task. The output for classification tasks is one of the Labels, while for Number the output is a string of numerical value.

The **input similarity** quantifies the n-gram similarity of the input corpuses of the two groups. Suppose that we sample $M$ inputs from the source and target groups respectively, denoted as $x_s = \{x_{s_1}, ..., x_{s_M}\}$ and $x_t = \{x_{t_1}, ..., x_{t_M}\}$, we calculate the Spearman's rank order correlation between the bag-of-word vectors of $x_s$ and $x_t$,

$$\rho = \frac{cov(V_s, V_t)}{\delta(V_s)\delta(V_t)}$$

where $V_s$ and $V_t$ denotes the ranked bag-of-word vectors of $x_s$ and $x_t$ on the vocabulary of $x_t$.

**Calculation of Q2 Metrics.** We sample the same amount of inputs from SocialIQA, PIQA and OpenbookQA, and denote the input corpuses as $x_1, x_2$ and $x_3$. Firstly, we calculate the **proportion of unique n-grams** for each group against the number of all n-grams for the three corpuses as

$$\frac{|\text{n-grams}(x_i)|}{|\text{n-grams}(\{x_1, x_2, x_3\})|}, i = 1, 2, 3$$

where n-gram($\cdot$) returns the set of unique n-grams, and the braces denotes mixing the inputs.

Secondly, we think the source group that has already covered a larger proportion of n-grams of the target group may promote better generalization, and we calculate the **proportion of n-gram coverage** between the source and target groups as

$$\frac{|\text{n-grams}(x_s) \cap \text{n-grams}(x_t)|}{|\text{n-grams}(x_t)|}$$

For both metrics, the n-gram($\cdot$) is calculated as both word 1-gram and character 4-gram using scikit-learn.

**Q1 Metrics for More Tasks.** Table 12 and Table 13 show the two Q1 metric results for commonsense QA and Dialog tasks. Linking the results with the generalization performance in Table 1 and Table 2, we have the following observations. 1) For each target group of the commonsense QA task, the largest value for input similarity coheres with the best generalization performance, but the smallest value of label distribution shifts does not correlate to the best generalization performance. 2) For the Dialog groups, the zero label distribution shifts and the close input similarities cohere with the subtle generalization performance difference on each target group. 3) The evaluation metrics cannot be compared across target groups nor across tasks. *e.g.,* the source group SocialIQA performs better on PIQA than OpenbookQA (*cf.* Table 2), but the input similarity is higher for OpenbookQA. Also, MuTual has smaller input similarity with Ubuntu (input similarity is 0.56, and generalization performance is 74.7) but better generalization performance than PIQA generalizing to SocialIQA (input similarity is 0.57, and generalization performance is 68.9) (*cf.* Section 2). These findings reveals the benefits and limitations of the Q1 metrics.

| Source \ Target | SocialIQA | PIQA | OpenbookQA |
|---|---|---|---|
| SocialIQA | - | **2.44** | **0.27** |
| PIQA | **0.38** | - | 0.59 |
| OpenbookQA | 1.59 | 3.17 | - |

(a) Commonsense QA

| Source \ Target | Mutual | DSTC7 | Ubuntu Dialog |
|---|---|---|---|
| Mutual | - | 0 | 0 |
| DSTC7 | 0 | - | 0 |
| Ubuntu Dialog | 0 | 0 | - |

(b) Dialog

Table 12: Results for label distribution shifts. Smaller value indicates smaller distribution shift. Bold font indicates the smallest value for each column.

| Source \ Target | SocialIQA | PIQA | OpenbookQA |
|---|---|---|---|
| SocialIQA | - | 0.59 | 0.62 |
| PIQA | 0.57 | - | **0.69** |
| OpenbookQA | **0.61** | **0.67** | - |

(a) Commonsense QA

| Source \ Target | MuTual | DSTC7 | Ubuntu Dialog |
|---|---|---|---|
| MuTual | - | 0.55 | 0.56 |
| DSTC7 | 0.56 | - | 0.56 |
| Ubuntu Dialog | 0.57 | 0.57 | - |

(b) Dialog

Table 13: Results for input similarity. Larger value indicates smaller distribution shifts. Bold font indicates the largest value for each column.

## A.4 Details for Baseline Implementation

For all compared methods, the LLM parameters such as temperature, top p, max tokens are the same as in Appendix A.2. The implementation and results for APE follow the preliminary experiments as illustrated in Appendix A.2 and Section 2. For APO, we follow the original parameter setting except for number of optimization step as 1 because the three tasks do not need multi-round optimization. For GPO, the value $K$ is unchanged from APE. The consistency threshold for GPO are 0.83 (5 out of 6 prompts) for sentiment analysis and commonsense QA, and 0.33 (2 out of 6 prompts) for DROP. Note that APE and APO are not designed to utilize the unlabeled target group data so we only observe the direct generalization performance, while APE-ut and GPO utilize the $N$-shot source group data and $N$-shot target group data. All of the above methods do not need to apply Monte Carlo search following the official implementation of APE. We use one 32GB GPU to perform inference for Vicuna models. We present the meta prompt of APE and APE-ut, the initial prompt for APO, the human-written prompts, the revised versions by PromptPerfect here.

- **APE meta prompt**:
  *I provide my friend with an instruction. Based on the instruction, I gave him several inputs, and he generated the corresponding outputs. Here are the input-output examples:[DEMO]. Please briefly illustrate the instruction and describe the output format. The instruction is to*

- **APE-ut meta prompt**:
  *I provide my friend with an instruction. Based on the instruction, I gave him several inputs, and he generated the corresponding outputs. Here are the input-output examples:[Source]. Here are also some unlabeled examples. Please consider these examples as well for prompt generation:[Unlabeled Target].Please briefly illustrate the instruction and describe the output format. The instruction is to*

- **APO initial Prompts**:
  For Yelp: *Provide a sentiment analysis of the following text. Answer Positive Neutral or Negative as labels.*
  For SocialIQA: *Give answer to the following multi choice question. Provide only the single letter as labels.*
  For Number: *Answer the following question based on the context which involves numerical calculation. Provide only the numerical value that directly answers the question.*

- **Human Prompts**:
  For sentiment analysis: *Provide a sentiment analysis of a given input text. The output format is a single word indicating whether the sentiment is positive, negative, or neutral.*
  For commonsense QA: *Give answer to the following multi choice question which involves commonsense knowledge. Provide only the single letter (a, b, c, or d) as labels.*
  For DROP: *Answer the following question based on the context which involves numerical reasoning. Provide only the direct answer the question, which can be a numerical value or a short string.*

- **PromptPerfect**:

  For sentiment analysis: *Your task is to perform a sentiment analysis on a given input text and provide a single word indicating whether the sentiment is positive, negative, or neutral. The input text may contain any language or style of writing. Please ensure that your analysis takes into account the overall tone and context of the text.Your response should be concise and clear, providing a single word that accurately reflects the sentiment of the input text. If there are multiple sentiments present in the text, please choose the one that best represents the overall feeling conveyed by the author.Please note that your analysis should take into account all relevant factors, such as tone, language use, and content. Your response should also be flexible enough to allow for various types of input texts.*

  For commonsense QA: *Please choose the best answer for the following multiple choice question. Choose the one answer that best fits the given scenario. Please provide only the single letter (a, b, c, or d) as labels.*

  For DROP: *Your task is to answer a numerical question based on a given context involving numerical reasoning. Please provide a direct answer to the question, which can be a numerical value or a short string.Please note that your response should be concise and directly answer the question. The question may involve various numerical data, such as percentages, averages, or counts. You should focus on identifying the relevant information and providing a clear and accurate answer.Additionally, please ensure that your response is flexible enough to allow for various relevant and creative answers based on the context provided.*

## A.5 Case Study

We present a case study by presenting the best prompt among the five runs for sentiment analysis and DROP as shown in Table 14. We can observe that the optimized prompt for a single group often contains group-specific background information as highlighted by underline which may hinder robust prompt generalization. On the contrary, the optimized prompts of GPO are more general and thus performs well on both groups. Note that for Spans, the optimized prompt is also general enough and thus can generalize well to Number as shown in

Table 2.

| Yelp | *Provide feedback on various experiences, such as dining, shopping, and service. The output format is a sentiment analysis, where the input is analyzed to determine whether the experience was positive, negative, or neutral. The output is a single word indicating the sentiment of the experience.* |
|---|---|
| Flipkart | *Provide a sentiment analysis of customer reviews. The input consists of a customer review of a product, and the output is a binary classification of the sentiment as either positive or negative.* |
| GPO | *provide a sentiment analysis of a given text. The output format is a single word indicating whether the sentiment is positive, negative, or neutral.* |
| Number | *Answer a specific question based on a given context. The output format is a numerical value that directly answers the question asked.* |
| Spans | *Answer a specific question based on a given context. The output format is a single word or phrase that directly answers the question asked.* |
| GPO | *Answer questions based on given context information. The output format is a numerical value or a single word answer.* |

Table 14: Case study on the prompts optimized by APE from a source group, and GPO.

| $K$ | Flipkart Ensemble |
|---|---|
| 3 | $81.2 \pm 1.3$ |
| 6 | $84.5 \pm 2.0$ |
| 9 | $85.8 \pm 1.9$ |
| 12 | $85.2 \pm 1.8$ |
| 18 | $85.3 \pm 1.4$ |

Table 15: Generalization performance of GPO on Flipkart with different numbers of candidate prompts $K$.

## A.6 Study on the Impact of the Number of Candidate Prompts

We examine the effect of varying the number of candidate prompts $K$ on GPO performance in our 36-shot sentiment analysis task. We test the $K$ values in $\{3, 6, 9, 12, 18\}$. The results on the target group Flipkart are shown in Table 15. We observe that the generalization performance stabilizes as $K$ reaches a specific value, in this case is 6, indicating that further generating more prompts are unlikely to yield significant improvements in performance.