# OpenReview forum: "Robust Prompt Optimization for Large Language Models Against Distribution Shifts"
_EMNLP/2023/Conference — EMNLP 2023 Main_

### Official Review · Reviewer_Uxx8 · 2023-08-04

**Soundness:** 3

**Excitement:**

3: Ambivalent: It has merits (e.g., it reports state-of-the-art results, the idea is nice), but there are key weaknesses (e.g., it describes incremental work), and it can significantly benefit from another round of revision. However, I won't object to accepting it if my co-reviewers champion it.

**Missing References:**

- Your solution for improving prompt robustness is similar to using any unlabeled data to augment the supervised examples for improving performance, although your unlabeled data is originating from a target group. The paper does not discuss the past literature of using unlabeled data in NLP tasks. At least such methods must be discussed as part of the related work section, although your noise reduction techniques are based on prompt ensemble and specific to Large LMs.

**Paper Topic And Main Contributions:**

- The paper studies the robustness of the prompts for LLMs optimized using a recent gradient-free method. The analysis shows large performance gaps across famous NLP datasets. The solution to incorporate the unlabeled data from the target domain while optimizing the prompts on the source data improves the robustness under domain shift.

**Questions For The Authors:**

- How do you generate multiple outputs from these LLMs? is it the API or a specific decoding algorithm?

**Reasons To Accept:**

- This is a well-written paper. The hypothesis and experiments are well defined and explained. The experimental analysis conducted by this paper over six different NLP tasks across 16 datasets clearly shows that with a recent gradient-free prompt optimization technique such as APE over a strong LLM such as GPT3.5, we may have a large performance gap if we optimize the prompts on the source group while testing on the another target group. This can form a standard benchmark for future research about the generalization and robustness of these prompt optimization techniques.

**Reasons To Reject:**

- The proposed solution to solve the robustness issue under domain shift is not very ML grounded and cannot be generically applied to smaller LM models such as BERT-large or T5-large. The Meta prompt you have used to generate task instructions to be used for labeling the unlabeled target group, may not be applicable to these medium size LMs.

**Reproducibility:**

4: Could mostly reproduce the results, but there may be some variation because of sample variance or minor variations in their interpretation of the protocol or method.

**Reviewer Confidence:**

4: Quite sure. I tried to check the important points carefully. It's unlikely, though conceivable, that I missed something that should affect my ratings.

**Typos Grammar Style And Presentation Improvements:**

- Line 047, 235 fix typo.

---

> ### Author Rebuttal · Authors · 2023-08-29
>
> Thank you for your positive feedback regarding the quality of the paper writing, the clarity of the problem definition and experiments, as well as the potential for future research.
>
> **Reason to Reject**
>
> Thanks for your perceptive comments. Firstly, our GPO with the meta prompt is tailored for black-box API LLMs such as ChatGPT which have garnered substantial attention recently. Their model parameters and gradients are not accessible, leading to gradient-free prompt optimization solutions such as APE and GPO. Secondly, for the medium-sized LMs, their model parameter and gradients are usually accessible. As such, there is no need to utilize these gradient-free prompt optimization methods. Instead, gradient-based approaches tend to yield better performance. In light of these, it is probably not necessary to develop a gradient-free method suitable for both black-box LLMs and medium-sized LMs.
>
> **Question**
>
> Thanks for your question. We use the API to generate the outputs. To further clarify, we do not sample multiple output for one input. Instead, we create multiple inputs with different in-context examples, and generate one prompt for each input.
>
> For the missing reference, thanks for pointing it out. We will discuss these related literature and revise the paper accordingly. We will also fix the typo.

---

### Official Review · Reviewer_P2wd · 2023-08-05

**Soundness:** 4

**Excitement:**

4: Strong: This paper deepens the understanding of some phenomenon or lowers the barriers to an existing research direction.

**Paper Topic And Main Contributions:**

- Existing prompt optimization techniques find the best prompt for the labeled data which doesn't necessarily generalize to test data with different distributions.The authors propose the GPO approach to overcome this.
- They propose a Generalized Prompt Optimization (GPO) technique that incorporates LM generated labels for a portion of the target data and optimizes on both the source and LM labeled target data.
- The GPO approach generates an ensemble of prompts based on examples, applies this ensemble to the target data, filters target examples based on prompt agreement and use these target examples mixed with source examples for prompt optimization.
- The authors show that GPO improves target distribution performance on different benchmark datasets.
- This approach has many great applications.

**Questions For The Authors:**

A. What is the effect of the type of in-context examples on the upper bound of the accuracy? Was that evaluated at any point. Since the work has many real world applications it would be interesting to analyze this effect to see if the examples can be optimized to get better upper bound accuracy. The model might also be sensitive to the order of these examples. Was any similar experiment conducted?
B. What are the parameters for generation like top p, temperature, frequency penalty and max tokens for the meta prompt and optimized prompt. What was the optimization approach for these parameters?

**Reasons To Accept:**

- This work identifies an important real world problem which can be deployed to production for multiple products with some great applications for difficult tasks.
- This work tackles the lack of robustness of prompt optimization to distribution shifts for different prompt engineering tasks.
- It is interesting how this works formulates the motivation and problem empirically.
- GPO uses LLM capabilities for labeling and prompt optimization.
- The authors demonstrate improve robustness to distributional shifts through extensive experiments across multiple datasets and tasks, over different baselines along with a detailed ablation study and analysis.
- The paper is well-written with clarity of technical details and easy to understand.

**Reasons To Reject:**

- Lack of comparison with other gradient-free prompt optimization techniques. What is the effect on robustness and upper bound of accuracy in comparison to baselines mentioned in the related works?

**Reproducibility:**

4: Could mostly reproduce the results, but there may be some variation because of sample variance or minor variations in their interpretation of the protocol or method.

**Reviewer Confidence:**

3: Pretty sure, but there's a chance I missed something. Although I have a good feel for this area in general, I did not carefully check the paper's details, e.g., the math, experimental design, or novelty.

---

> ### Author Rebuttal · Authors · 2023-08-29
>
> Thank you for your recognition of the importance of the problem, the extensiveness of the experiments, and the clarity of the paper writing.
>
> **Reason to Reject**
>
> Thanks for your great question. As illustrated in the related work, there are several types of gradient-free prompt optimization techniques. The rationale for selecting compared methods is explained as below:
> - Search-based methods: This category of methods is the current state-of-the-art on black-box API LLMs, where we adopt APE and APO as baselines.
> - RL-basaed methods: Comparison with RL-based methods was omitted due to their substantial underperformance in relation to APE and APO, as evidenced in the APO paper (Pryzant et al., 2023). We conjecture that RL-based methods might not be suitable for prompt optimization with limited data examples due to the limited reward signals.
> - Other gradient-free methods: Although some methods claim suitability for black-box LLMs, they are not entirely compatible for black-box API LLMs like ChatGPT. The evolutionary algorithm (Sun et al., 2022) needs to input prompt embeddings to the LLM, and the boosting method (Hou et al., 2022) requires passing gradients through the LLM. Prompt embeddings and passing gradient are unavailable for the black-box API LLMs.
> Therefore, we mainly focus on the state-of-the-art search-based methods for investigation.
>
> **Question A**
>
> Thanks for pointing out these insightful questions.
> - For the impact of the order of the in-context examples, we choose one data split of Yelp and Flipkart, and randomly shuffle the order of the in-context examples for five times. The averaged GPO performance is shown below.
>
>      | Dataset | Top 1| Ensemble|
>
>      | Flipkart | 80.6 $\pm$ 1.2| 80.9 $\pm$ 1.3|
>
>      | Yelp | 78.5 $\pm$ 0.6 | 79.0 $ \pm$ 0.9|
>
> We can observe that the variances are all small, showing that the GPO performance is not very sensitive to the order of the in-context examples. This is probably due to the small differences in candidate prompts generated by varying in-context example orders, thereby yielding small impact on performance disparities.
> - We have not yet conducted experiments on different types of in-context examples. To achieve this, it is essential to classify existing in-context examples into different types, which are not directly available. We will try to design classification strategies and investigate their effects in the future. Additionally, in line with your insightful idea, optimizing in-context examples is a promising direction, and we have earmarked it for our future endeavors (mentioned in the future work and limitations).
>
> **Question B**
>
> As illustrated in Appendix A.2, we set top p as 1.0, temperature as 0 to control the randomness of LLMs for better result replication. About the max tokens, for the input meta prompt with in-context examples, the length limit aligns with the LLM's input length limit, where truncation will not occur. For the generated optimized prompts, we set the max tokens as 100 to ensure no truncation of the generated prompts. For other parameters for generation, including the frequency penalty, we follow APE since APE is our base method with the same prompt generation and evaluation process.

---

### Official Review · Reviewer_XdbA · 2023-08-12

**Soundness:** 4

**Excitement:**

3: Ambivalent: It has merits (e.g., it reports state-of-the-art results, the idea is nice), but there are key weaknesses (e.g., it describes incremental work), and it can significantly benefit from another round of revision. However, I won't object to accepting it if my co-reviewers champion it.

**Paper Topic And Main Contributions:**

This study presents the Generalized Prompt Optimization (GPO) framework for LLMs, aiming to enhance generalization performance.

**Reasons To Accept:**

- The manuscript is well-structured and easily comprehensible.
- The experimental results substantiate the efficacy of GPO.
- This work examines the distribution shifts across diverse tasks.

**Reasons To Reject:**

- The baseline method lacks experiments that explore combinations with various LLMs.
- Certain hyperparameters seem to influence overall performance, such as the number of candidate prompts $K$, an aspect overlooked in the authors' analysis.

**Reproducibility:**

3: Could reproduce the results with some difficulty. The settings of parameters are underspecified or subjectively determined; the training/evaluation data are not widely available.

**Reviewer Confidence:**

3: Pretty sure, but there's a chance I missed something. Although I have a good feel for this area in general, I did not carefully check the paper's details, e.g., the math, experimental design, or novelty.

---

> ### Author Rebuttal · Authors · 2023-08-29
>
> Thank you for your positive feedback regarding the comprehensiveness of the paper and the substantiality of the results.
>
> **Reason to Reject 1**
>
> In Table 9, we have explored combinations with GPT-3.5, Vicuna 7B, and Vicuna 13B, by comparing APE, the best baseline on the sentiment analysis dataset Flipkart, and GPO on Flipkart. In addition, we do new experiments with GPT-4. We will add the new results to our paper in the next version. Here we summarize the results of combining with various LLMs on Flipkart.
>
> | LLM  | APE ensemble | GPO ensemble |
>
> | Vicuna 7B | 43.9 $\pm$ 21.3 | 71.9 $\pm$ 13.1 |
>
> | Vicuna 13B | 68.3 $\pm$ 13.7 | 70.7 $\pm$ 10.8 |
>
> | GPT-3.5 | 80.5 $\pm$ 2.1 | 84.5 $\pm$ 2.0 |
>
> | GPT-4 | 83.3 $\pm$ 0.0 | 85.4 $\pm$ 2.4 |
>
> From the table, we can observe that 1) the performance gap between APE and GPO exists across LLMs, even for the state-of-the-art GPT-4, showing that the generalization issue exists across LLMs; 2) GPO largely outperforms APE, showing its effectiveness.
>
> **Reason to Reject 2**
>
> Thanks for your insightful comments. As illustrated in Appendix A.2, our approach divides the N-shot data for prompt generation into K splits, where each split generates one candidate prompt, thus we experiment with different numbers of candidate prompts K in the range of N's factors. For example, on the 36-shot sentiment analysis task, we show the performance on Flipkart with the values of K in {3, 6, 9, 12, 18}.
>
> | values of K | 3 | 6 | 9 | 12 | 18 |
>
> | GPO ensemble | 81.2 $\pm$ 1.3 | 84.5 $\pm$ 2.0 | 85.8 $\pm$ 1.9 | 85.2 $\pm$ 1.8 | 85.3 $\pm$ 1.4 |
>
> From the above table, we can observe that the performance stabilizes as K reaches a specific number, in this case, 6. This finding suggests that, beyond this point, further increases in K would not significantly enhance performance. We will add more analysis on hyperparameters into the Appendix.

---

### Meta-Review · Area_Chair_jbN9 · 2023-09-22

**Recommendation:** 5

**Metareview:**

In light of the significant recognition garnered by LLMs for their exceptional performance across various Natural Language Processing tasks, the exploration of prompt optimization emerges as a highly promising avenue for research.
This holds particular significance in the context of black-box LLMs such as ChatGPT.
This paper introduces a gradient-free prompt optimization technique, characterized by its robustness in mitigating the challenges posed by data domain shifts.
All reviewers have agreed that the proposed Generalized Prompt Optimization (GPO) framework is useful for improving generalization, especially under domain shifts.
While some concerns have also been raised that the comparison is a bit weak.
And during the rebuttal period, the authors have made efforts to address the concerns.
Generally, the merits of this paper outweigh its flaws.

---

### Decision · Program_Chairs · 2023-10-07

**Decision:**

Accept-Main

**Comment:**

In light of the significant recognition garnered by LLMs for their exceptional performance across various Natural Language Processing tasks, the exploration of prompt optimization emerges as a highly promising avenue for research.
This holds particular significance in the context of black-box LLMs such as ChatGPT.
This paper introduces a gradient-free prompt optimization technique, characterized by its robustness in mitigating the challenges posed by data domain shifts.
All reviewers have agreed that the proposed Generalized Prompt Optimization (GPO) framework is useful for improving generalization, especially under domain shifts.
While some concerns have also been raised that the comparison is a bit weak.
And during the rebuttal period, the authors have made efforts to address the concerns.
Generally, the merits of this paper outweigh its flaws.